# Alarmins as a Possible Target of Future Therapies for Atrial Fibrillation

**DOI:** 10.3390/ijms232415946

**Published:** 2022-12-15

**Authors:** Egidio Imbalzano, Giuseppe Murdaca, Luana Orlando, Marianna Gigliotti-De Fazio, Dario Terranova, Alessandro Tonacci, Sebastiano Gangemi

**Affiliations:** 1Department of Clinical and Experimental Medicine, University of Messina, n. Viale Benedetto XV, n. 6, 98125 Messina, Italy; 2Department of Internal Medicine, Ospedale Policlinico San Martino, University of Genova, 16132 Genova, Italy; 3Clinical Physiology Institute, National Research Council of Italy (IFC-CNR), 56124 Pisa, Italy; 4Department of Clinical and Experimental Medicine, School and Operative Unit of Allergy and Clinical Immunology, University of Messina, 98125 Messina, Italy

**Keywords:** alarmins, S100 protein, HMGB1, heat shock proteins, atrial fibrillation

## Abstract

To date, worldwide, atrial fibrillation is the most common cardiovascular disease in adults, with a prevalence of 2% to 4%. The trigger of the pathophysiological mechanism of arrhythmia includes several factors that sustain and exacerbate the disease. Ectopic electrical conductivity, associated with the resulting atrial mechanical dysfunction, atrial remodeling, and fibrosis, promotes hypo-contractility and blood stasis, involving micro endothelial damage. This causes a significant local inflammatory reaction that feeds and sustains the arrhythmia. In our literature review, we evaluate the role of HMGB1 proteins, heat shock proteins, and S100 in the pathophysiology of atrial fibrillation, offering suggestions for possible new therapeutic strategies. We selected scientific publications on the specific topics “alarmins” and “atrial fibrillation” from PubMed. The nonsystematic review confirms the pivotal role of molecules such as S100 proteins, high-mobility group box-1, and heat shock proteins in the molecular pattern of atrial fibrillation. These results could be considered for new therapeutic opportunities, including inhibition of oxidative stress, evaluation of new anticoagulant drugs with novel therapeutic targets, molecular and genetic studies, and consideration of these alarmins as predictive or prognostic biomarkers of disease onset and severity.

## 1. Introduction

Atrial fibrillation (AF) is the most common cardiac arrhythmia in the world, with an estimated prevalence of 2 to 4% [1]. It is defined as supraventricular tachyarrhythmia with uncoordinated atrial electrical activation, which results in chaotic and irregular activation of atrial pacemaker cells and thus ineffective mechanical contraction of the atrial chambers [2]. Advanced age is one of the independent risk factors, correlated with multiple disorders such as diabetes, hypertension, heart failure, obesity, and obstructive sleep apnea syndrome (OSAS) [3]. Because of the aging of the global population, it is expected to be more prevalent in individuals aged over 65 by 2060 [4,5]. The interaction of distinct factors, such as inflammation, structural remodeling, fibrosis, and ion-channel dysfunction, guarantees the onset of a complex pathophysiological mechanism that starts atrial fibrillation and worsens structural and electrical changes in the atria [2]. The generation of quick multiple ectopic electrical pulses is able to start and sustain irregular electrical activity of atrial fibrillation. The most common locations of occurrence of ectopic foci are the pulmonary veins, whose isolation is the cornerstone of catheter ablation procedures [6,7], and less commonly the interatrial septum, coronary sinus, and superior vena cava [8,9]. These electrical alterations also help the AF-associated hypercoagulable state. Failed electrical regularity and excessive ectopic contractility antagonize local atrial hypo contractility, increasing endothelial expression of plasminogen activator inhibitor (PAI-1) [10], contributing to clot generation. Various diseases are risk factors for the occurrence of AF, such as hypertension [11], heart failure [12] and diabetes [13,14]. Their common factor is represented by the inflammatory state involved in each of these disorders, which plays a key role in the pathophysiology of AF by mediating the production of cytokines and reactive oxygen species that can increase the disease state, fibrosis, and atrial remodeling [15]. Hypo contractility, as well as blood stasis, promotes the development of endothelial microdamage, which attracts the migration and infiltration of several cells of the innate immune system, including macrophages and leukocytes. This mechanism damages the atrial architecture, promoting an inflammatory process, remodels the walls of the atrium, fostering fibrous tissue growth and destroying cardiomyocytes, thus increasing the local inflammatory response and helping the expression of endothelial adhesion molecules and inflammatory cytokines [16,17]. Alarmins represent a group of endogenous molecules characterized by multiple functions. They can be classified into three categories: (1) granule-derived, as α- and β-defensins, cathelicidin (LL37/cathelicidin-related antimicrobial peptide (CRAMP), eosinophil-derived neurotoxin (EDN) and granulysin; (2) nuclear form, including HMGB-1, HMGN1, IL-33, and IL-1α; (3) cytoplasmic, as heat shock proteins (HSP-60, -70, -90, and -96), S100 proteins, ATP and uric acid [18]. These intracellular proteins are generally released as inflammatory signal mediators and represent the first defense against infections, as well as during trauma and various metabolic, physical or chemical injuries [19,20]. Their release can attract additional inflammatory molecules, such as leukocytes, triggering a massive immune local response [21] and activating dendritic cells. Roh et al. [22] reported that alarmins and various damage-associated molecular patterns (DAMPS), such as HMGB1, S100 and HSP-70, play a key role in the pathogenesis of inflammatory diseases. Several studies have shown a key role for HMGB1, heat shock proteins and s100 proteins in AF physiopathology. In our review, we showed how alarmins are very often involved in the inflammatory mechanism underlying AF disease. In this process, the involvement of various inflammatory markers is established in the literature [23,24]. Taken this into account, we performed a non-systematic review about the role of HMGB1, heat shock proteins and S100 alarmins in the pathophysiological mechanisms of AF, hypothesizing their role as potential novel therapeutic targets for arrhythmia.

## 2. Methods

The search was conducted on PubMed and limited to the articles published until 15 June 2022. We used the following MeSH terms: “S100”, “HMGB1”, “heat shock proteins”, “IL-33”, “IL-1a”, “defensins”, “ropocamptide”, and “atrial fibrillation”. We selected articles by title and abstract; the entire article was read in case the title/abstract indicated that the association between atrial fibrillation and alarmins had been evaluated. We decided to exclude the following alarmins because we found no results: IL-1α, IL-33, ropocamptide, and defensins. Articles were included in our review based on the following inclusion criteria: English language, publication in peer-reviewed journals. Articles were excluded according to the title, abstract or full text for irrelevance to the topic at hand. Other exclusion criteria were review articles, meta-analyses, editorial comments, case reports, and non-English articles. Three authors (L.O., M.G., D.T.) conducted the initial search and separate review, selecting articles according to the inclusion and exclusion criteria. Finally, we included studies dealing with molecular, animal and human targets.

## 3. Results and Discussion

Overall, 31 articles dealt with the association between AF and HMGB1, S100 and heat shock proteins, as displayed in Table 1, Table 2 and Table 3, respectively.

### 3.1. HMGB-1 and Human Studies

High-mobility group box-1 (HMGB1) protein, also called amphoterin, is an alarmin named for its high electrophoretic mobility. HMGB1, encoded by a gene located on chromosome 13q12, is a non-histone structural chromatin-binding protein with 215 amino acids and a molecular weight of 25 kDa. The protein contains two homologous proximal DNA-binding domains, called A-box (9–79 aa) and B-box (95–163 aa), and a C-terminal acidic tail (186–215 aa) with repeated glutamic and aspartic acid residues. Two nuclear localization signals (NLS1 and NLS2) and two nuclear export signals (NES) drive the translocation of HMGB1 from the nucleus to the cytoplasm. The localization of HMGB1 is pivotal for its function and plays a key role in acute and chronic inflammation, with effects in several diseases [54]. HMGB1 is located in the nucleus, cytosol, or extracellular space, where it is passively released from lytic cell death or actively secreted by viable cells. Under physiological conditions, HMGB1 is anchored in the nucleus. When HMGB1 is released, it modulates cellular stress responses and inflammation [55]. Several studies confirmed a key role of HMGB1 in AF (Figure 1). Qu et al. showed the higher incidence of postoperative AF in patients with rs2249825 polymorphism of HMGB1 protein receiving elective cardiac surgery (CABG). High levels of HMGB1 protein were related with genotype CG + GG versus genotype CC [56]. Recent onset postoperative AF is one of the most common complications after cardiac surgery [31]. The injury triggers a local inflammatory response that results in elevated serum concentrations of inflammatory biomarkers. This could represent a close correlation between oxidative stress and HMGB1; as the first increases, so does the second one. Such a concept would open possibilities for anti-AF therapies accurately targeting the inflammatory state underlying the pathology. Higher concentrations of tissue factor (TF) and HMGB1 protein were found in the left atrial appendage (LAA) tissue of 45 patients with AF and thrombosis receiving surgical valve replacement for rheumatic disease, compared with patients without thrombosis. This study showed a linear correlation of HMGB1 with TF, myeloid differentiation factor 88 (MyD88) and nuclear factor kB (NfkB) [57]. It was also shown that high levels of oxidative stress proteins, such as malondialdehyde and hsCRP, are positively correlated with HMGB1 levels in AF patients. HMGB1 protein, in association with TF, plays a key role in the downstream regulation of MyD88/NfkB, suggesting that HMGB1 could cause thrombosis through the MyD88/NfkB pathway. The relationship between HMGB1 and thrombogenesis is carried out through the RAGE, TLR4 and MyD88 pathways, with activation of platelets (TLR-4, TLR-2), determining thrombosis in LAA. Another study suggests evidence for immune-mediated platelet activation (TRL-2, TRL-4 and HMGB1 protein) in the left atria of patients with AF, in which Toll-like receptor 2 and Toll-like receptor 4 were higher in persistent AF, which could suggest the hypothesis that the role of HMGB1 in atrial thrombogenesis in AF patients could be via TLR-2 and TRL-4 [32]. As is well known, the cardio-embolic risk caused by AF is the basis of the therapeutic pathway with the aim of reducing stroke incidence [33]. While macroscopically, atrial noncompliance, atrial remodeling and fibrosis, and blood stasis may be causes, HMGB1 appears to independently control high thrombogenic activity in patients with AF and plays a key role in the thrombogenic process. This suggests a new role for HMGB1 in thrombogenesis in AF patients and new potential goals for future anticoagulant therapies. Increasing the serum concentration of HMGB1 was achieved in 86 patients with paroxysmal AF and persistent AF. The serum concentration of HMGB1 was higher in persistent AF patients [58]. HMGB1 was discovered to have various functions within a molecular pathway. When it resides outside the cell, it is one of the most powerful signals of inflammation, mediated by the innate immune system; when it resides in the nucleus, HMGB1 binds and folds the DNA helix, with the purpose of protein formation and ensuring nuclear biochemical processes [30]. In our case, it appeared to be highly represented in patients with AF, correlating its concentration with that of MDA and HS-CRP, which represents another inflammatory protein. HMGB1 could be the amplifier of the inflammatory response with the recall of proinflammatory cytokines such as IL-1, tumor necrosis factor (TNF) and IL-6, which handle atrial remodeling and fibrosis, as well as in the onset of post-surgical AF. Moreover, exactly where the inflammatory mechanism is continuous and constant, as in permanent AF, it seems to occur in higher concentrations than in the paroxysmal AF condition, as an independent predictor of AF.

### 3.2. S100 Protein and Human Studies

In this review, we also analyzed the role of the S100 proteins. Generally, S100s are calcium-binding proteins and modulators of various enzymatic activities, including differentiation, inflammation, proliferation, migration, apoptosis, Ca^2+^ homeostasis, and energy metabolism [59,60]. S100 proteins are a subfamily of EF-hand, calcium-binding proteins with integrated dimeric structure that can form higher-order oligomers [61]. They represent some of the major second messenger transducers of intracellular Ca^2+^ signals. They also play a significant role in the extracellular environment, which is pivotal in the innate immune response and activation of inflammation. Out of the 25 different S100 genes, only S100A7, S100A8, S100A9 and S100A12 have been identified as modulators of innate immunity. The family of S100 proteins is exceptionally large; to date, it appears that S100A4 and S100β are related to AF in the literature (Figure 2). Scherschel et al. showed higher concentrations of S100β protein in AF patients undergoing catheter ablation. Neuronal injury by the intrinsic cardiac autonomic nervous system (ICSN) upon catheter treatment released damage-associated molecular pattern proteins (DAMPs) as S100β protein. In vitro studies dealing with murine intracardiac neurons showed that S100β decreased potential action and increased neuronal cell growth [62]. This confirms its well-known role as an inflammatory biomarker. At the same time, the trophic role of S100β was also shown, which, released from glial cells through the RAGE receptor, would be able to regenerate peripheral cardiac nerves. Moreover, at 6 months after surgery, patients had fewer recurrences of AF. This would demonstrate a high protective role of the molecule in the onset of AF and could reveal its crucial role as a predictor marker of disease. Kato et al. showed that, in the atrial tissue of AF patients (paroxysmal AF, persistent AF) undergoing left atrial appendectomy during cardiac surgery, extension of atrial fibrosis was related with the amount of S100A4 protein [27]. Another novel therapeutic approach to prevent atrial remodeling could be to evaluate S100A4 protein, known as fibroblast specific protein 1 (FSP1), related to endothelial–mesenchymal transition of endothelium atrial cells and with the amount of fibrosis and atrial size. It was demonstrated that high concentrations of this molecule positively correlated with fibrotic depositions in the left atrium and atrial size. [30]. Scherschel et al. [26] showed high levels of S100β protein in paroxysmal AF patients undergoing pulmonary vein isolation (PVI) with different techniques, either radiofrequency (RF) or cryoballoon (CB), confirming its role as a biomarker of neurological damage. A study of 243 patients showed that AF patients had high levels of cerebral injury-related circulation proteins, such as TAU protein, astrocyte-specific glial acidic fibrillary protein (GFAP), and growth differential factor 15 (GDF15), but no calcium-binding protein B S100β [29]. Sramko et al. [28] evaluated S100B concentrations in patients with atrial fibrillation after catheter ablation who underwent brain MRI before and after the procedure. They investigated whether detection of ablation-related brain damage could be improved by assessment of S100B protein, a biomarker of brain damage. Very high values were recorded immediately after the procedure in patients with permanent atrial fibrillation; moreover, they appeared to be directly related to atrial size. The analysis of these two studies led to the evaluation of S100β as a biomarker that is extremely sensitive to cardiac damage but not specific to brain damage. More importantly, this protein is an early marker of blood–brain barrier opening that may precede neuronal damage, possibly influencing future therapeutic strategies. Furthermore, high concentrations of S100β are indicators of recent brain damage and predictors of adverse pathology outcome or possible diagnostic means to differentiate extensive from minor and transient damage [25].

### 3.3. Heat Shock Proteins and Human Studies

Heat shock proteins (HSPs) are molecular chaperones that are key to the preservation of cellular functions by preventing misfolding and aggregation of polypeptides and facilitating protein folding. The main function of HSPs is to serve as molecular chaperones, which are necessary for the folding of newly synthesized proteins and for the protection of proteins during exposure to stressful situations, including heat shock. Different cellular sites, such as the cytosol, endoplasmic reticulum, and mitochondria, have specific HSPs to meet their requirements [63]. In our review of the literature, heat shock proteins were shown to be the most studied proteins in the field of AF. We analyzed 11 human studies, eight animal studies, and two cell studies. In this section, we will show the results obtained on humans. Mandal et al. [64] recruited 80 patients undergoing elective coronary artery bypass surgery and showed that intracellular, but not serum, HSP70 levels were negatively correlated with post-operative AF, showing higher concentrations in patients not developing arrhythmia. With the aim of understanding whether the development of postoperative AF was related to different concentrations of atrial myocardium HSP70, Rammos et al. [37] demonstrated that patients with low preoperative HSP70 levels in the atrial myocardium had a significantly higher incidence of postoperative AFIB. In addition, patients without rhythm disturbances had significantly higher preoperative cellular HSP70 expression than patients with AFIB. However, HSP70 presence did not correlate with the time of AFIB onset, nor the duration or the resistance of AFIB reversal to different administered medications. Underlying this result would be the protective antiarrhythmic role of the molecule. This is explained by the fact that intracellular HSP70 and serum HSP70 play separate roles. It would have a protective role only inside the cell, losing this function when released into the serum. Demidov et al. [35] confirmed the cardioprotective role of HSP70 in humans, examining correlations between HSP70 levels in the patients’ hearts and the blood markers of cardiomyocyte alteration. Conversely, concentrations of soluble HSP70 may be a marker of stress-induced cellular damage, not correlating with intracellular levels. Pizon et al. [65], comparing the plasma HSP70 concentrations in eight men undergoing procedures with CPB (CABG group, coronary artery bypass) and eight men undergoing off-pump surgery (OPCAB group), showed that cardiopulmonary bypass (CPB) leads to an increase in circulating HSP70. In the CABG group, a gradual, continuous increase in the plasma concentration of HSP70 was observed during surgery, with the maximum peak 1 h after surgery, in clear contrast with the OPCAB group, in which a small, but not statistically significant, increase in HSP70 was found 1 h after surgery. In both groups, higher postoperative values were found for circulating HSP70 among patients with AF compared with the non-AF group. In line with earlier reports, Rigopoulos et al. [66] demonstrated a correlation between circulating serum HSP70 and the inflammatory cytokines serum interleukin-2 (sIL-2) and serum interleukin-4 (sIL-4) (Figure 3) with cardioversion and AF recurrence within 1 year. Among 90 patients with ROAF and 30 controls, it was found on the one hand that sHSP70 and sIL-2 were higher in patients with AF, while on the other hand, sIL-4 showed lower concentrations than in controls. Moreover, patients who did not undergo cardioversion showed high levels of inflammatory cytokines. From this, we conclude that serum HSP70 may have a key role as a prognostic marker of AF recurrence. A prognostic role in AF development can also be suggested for serum anti-HSP70 concentration. Among 45 patients admitted to the hospital for elective coronary artery bypass graft (CABG) surgery, Oc et al. demonstrated the association of circulating anti-HSP70 antibodies before and after surgery with postoperative AF [67]. Kornej et al. [42], with the aim of analyzing the association between HSP70, anti-HSP70 antibodies and their changes after catheter ablation of AF, found that anti-HSP70 antibody levels were associated with the type of AF; patients with persistent AF had higher anti-HSP70 antibody titers than their counterparts with paroxysmal AF, and both HSP70 and anti-HSP70 increased after ablation injury. In addition, longer ablation times with higher energies would correspond to an increase in both molecules. Furthermore, the increase in HSP70 and anti-HSP70 antibodies was associated with higher frequency of AF recurrence 6 months after the procedure. These results lead to the hypothesis of a possible, crucial role not only of inflammation, but also of autoimmunity, in the development of AF, triggering the question: what if AF could be an autoimmune disease? This is an intriguing observation already posed in the literature, where evidence would explain the occurrence of this kind of arrhythmia in patients with autoimmune diseases (Graves’ disease, rheumatic diseases, etc.) [44]. A high concentration of HSP70 represented a cellular stress response and an adaptive reaction to chaotic muscle activity [68,69]. It was shown that mortalin, mitochondrial heat shock protein 70 (mhsp70), an essential component of the mitochondrial import machinery, was 2.19-fold increased in its content in the atrial myocardium of patients with chronic AF, compared with patients with sinus rhythm. Afzal et al. [36] showed that the presence of methionine at the position 493 (493Thr) substitution in the HSP 70-Hom gene, coding for the HSP70 protein, located on chromosome 6p21.3 [43], was correlated with a higher incidence of postoperative AF, independent of other risk factors. Allende et al. [70] performed a microarray analysis in the peripheral blood cells of eight AF and stroke patients and eight AF subjects without stroke, discovering a stroke-related gene expression pattern. They found that HSPA1B, which encodes for the 70 kDa protein (Hsp70), resulted from a down-regulated gene in stroke individuals, thus reaching the definitive conclusion that down-regulation of Hsp70 actively plays a role in the cardio-embolic stroke pathogenesis. These results, in agreement with earlier findings, suggest a cytoprotective and antiarrhythmic role of the HSP70 protein. More studies could be conducted on the genetic mechanisms underlying AF, thus facilitating new management strategies for these patients. In addition, the involvement of these molecules also in hypercoagulability opens up possible therapeutic scenarios for new targets of anticoagulant therapies. In agreement, in a sub-analysis of the RIP-Heart study [49], it was decided to evaluate, through a genome-wide association study (GWAS), the correlation between genetic variants and the risk of occurrence of postoperative cardiologic diseases, including AF. The study, conducted on 1170 patients, showed that single nucleotide polymorphisms (SNP) of some loci, including that of HSPA8, are associated with new-onset AF in the postoperative setting. Other heat shock proteins have been evaluated in the literature. Marion et al. [52] evaluated the relationship between serum HSP levels and the presence of AF, stage of AF, and AF recurrence after electro cardioversion (ECV) or pulmonary vein isolation (PVI). Serum HSP-27, HSP-70, cvHSP and HSP-60 levels did not differentiate between AF stages and controls in sinus rhythm. However, HSP27 levels were increased during follow-up in patients with AF recurrence after ablative therapy, and this may lead to consideration of this marker as a predictor of disease. Finally, Brundel et al. [53] found a protective role of HSP27 in AF, demonstrating that upregulation of this protein by an induced injury, such as mild heat shock or by the drug geranylgeranyl acetone (GGA), protectively attenuates the induced myolysis and thus prevents electrical and structural remodeling. Moreover, HSP27 expression levels seem to be inversely correlated with the duration of arrhythmia and the extent of myolysis in paroxysmal and persistent AF.

### 3.4. Heat Shock Proteins and Animal Studies

According to our literature search, animal studies provided contradictory results on the supposed protective role of HSP70 in AF. Extensive evidence confirms that ischemic heart disease is a major trigger for the onset of AF [38,71,72]. In this regard, Sakabe et al. [73] demonstrated that overexpression of HSP70 has a protective role in the development of AF. In that study, they analyzed the effects of geranylgeranylacetone (GGA), an orally active inducer of HSP, on the development of AF on the substrate of atrial fibrillation associated with acute atrial ischemia in four groups of mongrel dogs. The authors thus demonstrated that GGA prevents ischemia-induced atrial conduction abnormalities and suppresses AF, and thus suggested that HSP induction could be a useful new anti-AF intervention for patients with coronary artery disease. Cross-referencing these two variables, it was observed that GGA-treated dogs were more resistant to the development of AF in both modalities, independently of undergoing atrial ischemia or not. This is related to the demonstrated overexpression of HSP70, which appears to have a protective role against ventricular myocardial ischemic insults and against AF. These findings would suggest that orally administered agents that induce the expression of HSP may be protective against some forms of AF in patients with coronary artery disease. In contrast to the previously mentioned study, a study by Vitadello et al. [39] did not find a cardiac protective role for HSP70 in fibrillating goat hearts. The authors observed that GRP94 levels were elevated in fibrillating goat atrial myocytes compared with normal atria and returned to control levels in atrial myocardium from cardioverted goats. Wakisaka et al. [34] confirmed the protective role of heat shock protein 72 (HSP72). They showed that overexpression of HSP72 induced by hyperthermia has a protective role against AF both in cell and in vivo. This was performed using Angiotensin II in rat fibroblasts and cardiomyocytes, knowing the profibrotic effects responsible for the development of AF. With the use of hyperthermia, they achieved overexpression of HSP72 and observed that the incidence of arrhythmia was lower where HSP72 was elevated. To confirm, they used a siRNA targeted to HSP72 in cells subjected to angiotensin II, and, with the reduction in HSP72 levels, a decrease in the antifibrotic effect was seen. In vivo, the use of repeated hyperthermia led to induction of HSP72 expression, resulting in attenuation of induced left atrial fibrosis and playing a key role in preventing atrial fibrosis and AF where Angiotensin II was involved. In their study, Li et al. [40] analyzed, on mouse cells, the kv 1.5 channel, which defines ultra-rapid delayed-rectifier potassium current (IKur), encoded by the KCNA5 gene, and Carboxyl-terminus heat shock cognate 70-interacting protein CHIP, E3 ubiquitin ligase composed of several components including HSC70. CHIP protein regulates the level of Kv1.5 protein and potassium channel function. The conclusions were that HSC70, which is an important part of CHIP, when downregulated, allows a higher incidence of AF than HSP70, the presence of which seems to be correlated with a lower incidence of AF, and this allows us to state that the expression of HSP70 seems to have a protective role in the development of AF in animals, as well. Completely contrasting hypotheses to the protective role of HSP70 in the literature have been advanced by Sapra et al. [47]. In a mouse model with HF and thus susceptible to AF, they studied the therapeutic potential of BGP-15, a hydroxamic acid derivative, which, when administered orally, induces HSP70 expression. It was observed that treatment with BGP-15 attenuated the increase in atrial size and lung weight, resulting in improved cardiac function, left ventricular size and systolic function, reduced cardiac fibrosis and collagen deposition; moreover, treatment with BGP-15 for 4 weeks was able to prevent or reduce arrhythmia episodes in the cohort of treated transgenic mice compared with the control group of transgenic mice that had not received it. However, surprisingly, these benefits did not come from the induction of HSP70, which was actually not increased in concentration, but from insulin-like growth factor receptor 1 (IGF1R). Thus, the focus was shifted to IGF1R and its expression, because an increase was seen in BGP-15-treated mice, while it was also observed that even in the absence of HSP70, cardiac fibrosis, molecular markers associated with cardiac pathology, and collagen deposition, were reduced. In light of this, the study suggested that HSP70 does not play a role in protecting against the development of atrial fibrosis, and this contradicts the studies seen to date. The study of AF extends to the nuclear pore complex, thanks to Zhang et al. [45], who for the first time identified the mutation in the NUP155 gene, which encodes a member of the nucleoporins—components of the nuclear pore complex (NPC)—as the cause of arrhythmia onset. Homozygous NUP155/mice die earlier than heterozygotes with the AF phenotype. This mutation is associated with inhibition of HSP70 mRNA export and nuclear import of HSP70 protein. Thus, although indirectly, the protective role of HSP-70 was confirmed. The genetic basis of AF could thus provide insights for new therapeutic and diagnostic approaches. In another study, the authors showed the protective role of HSP70 protein and atrial fibrillation in four groups of mice with cardiomyopathy (CM). HSP70 protein was higher in acute and stressful conditions such as CM, ischemia and induced atrial fibrillation. In particular, HSP70 was higher in the group with both CM and AF. However, a statistical correlation between AF inducibility rates and HSP70 in CM was not found in this study because AF in CM is induced by fibrosis involved in atrial remodeling [41]. In fact, Bernardo et al. [46] proved that chronically higher levels of HSP70 in a transgenic mouse model with heart failure and AF due to overexpression of muscle-restricted coiled-coil (MURC) did not improve reverse cardiac remodeling, fibrosis, and episodes of arrhythmia. Therefore, the overexpression of HSP70 had a protective role in stress condition such as ischemia and AF but showed no benefit in chronic cardiac disease.

### 3.5. Heat Shock Proteins and Cell Studies

Autophagy appears to be an important mechanism underlying atrial remodeling in AF. Wiersma et al. [48] proved that autophagy-induced endoplasmic reticulum (ER) stress perpetuates AF in a culture of HL-1 atrial cardiomyocytes, in Drosophila, in dogs, and in atrial biopsy specimens from patients with AF. The fascinating clinical implication of the discovery is that blockade of ER stress, by the chemical chaperone 4-phenylbutyrate (4PBA) and overexpression of the ER chaperone HSPA5, has been shown to inhibit activation of autophagy and thereby block electrical dysfunction and next atrial remodeling, in many in vitro and in vivo AF models. This may be a new therapeutic strategy, with a new molecular target, to limit the progression of AF, whereas, although indirectly, heat shock proteins cover a protective role for the disease. This is even more interesting when one considers that the molecular basis of many diseases triggering AF, such as diabetes mellitus, ischemic heart disease, and hypertension, derives precisely from endoplasmic reticulum stress [51,74,75,76]. Thus, a pleiotropic role of the 4-phenylbutyrate inhibitor, already approved for clinical use in urea cycle disorders, could be hypothesized. Han et al. [77] showed that laminin A/C nuclear mutation was associated with AF. The authors extracted the LMNA gene in a group of 610 patients with AF. They showed that the LMNA p.Arg399Csy mutation could be involved in the pathogenesis of AF. Nevertheless, p.Arg399csy mutation compromised the integrity of the nucleus under stress. Lamin A mutation replacement interaction of nucleoporin NUP 155, whose mutation is associated with familial AF, as described above. This weak linkage between p.Arg399Cys Lamin A mutation and NUP 155 is involved in alteration of HSP70 mRNA export and HSP70 protein import. In fact, immunostaining showed that the level of HSP70 was lower. Finally, the authors showed that LMNA mutation reduced a sodium current with alteration of the non-ion-channel gene. This complex mechanism could be involved in another pathogenesis of AF.

## 4. Conclusions and Future Perspectives

In summary, the link between alarmins and the pathophysiological mechanism of atrial fibrillation is clear and well known in the literature. The next level to be addressed could consider HMGB1, heat shock proteins and S100 proteins as new therapeutic targets for future medicine. We have seen an almost direct correlation between HMGB1 and oxidative stress in the inflammatory process of AF. This could allow HMGB1 to be evaluated as a possible diagnostic and prognostic biomarker of tissue damage. Wu et al. observed that HMGB1 is higher in patients with permanent AF, because they experienced a continuous inflammatory tissue damaging impulse. Therefore, we can consider HMGB1 as a prognostic factor of arrhythmia. The greatest risk of AF is stroke. From early studies, management of anticoagulation therapy has been the cornerstone along with rate control therapy [50,78]. There has been a shift, except for some conditions, from vitamin K inhibitors to new oral anticoagulants targeting factor X (rivaroxaban, apixaban, edoxaban) and factor IIa (dabigatran), each with their respective dosages [2,79]. To date, the literature has shown a prothrombotic and pro-coagulative role of HMGB1, increased in patients with AF and atrial thrombosis. Therefore, another therapeutic approach could consist precisely in an upcoming HMGB1-blocking anticoagulant therapy. Research on S100β was also in line with what was already known about the molecule. S100β appears to be increased in circulation in patients with AF after ablation injury or after pulmonary vein isolation surgery. Moreover, its behavior seems to be inversely proportional to recurrence. It would be useful to evaluate its use as an inflammatory biomarker of disease state. Furthermore, hypothesizing the protective role against recurrence, it could also be considered as a possible prognostic marker. This protein is an early marker of blood–brain barrier opening that may precede neuronal damage; indeed, it has been shown to be extremely sensitive to cardiac damage but not very specific to brain damage. In addition, high concentrations of S100β are indicators of recent brain damage. Thus, it becomes explicable as a predictor of adverse pathological outcomes or as a possible diagnostic means to differentiate extensive from minor and transient damage. This is a basis for the hypothesis of influencing future therapeutic and diagnostic strategies, such as evaluating its concentration as a possible indicator of brain damage, although not specific, and thus becoming a key point in risk stratification of patients. Finally, there is no shortage of the role of S100A4 as a predictor of disease state, the concentration of which seems to correlate with the state of fibrosis of the left atrium. In this case, one could then consider the molecule as a direct indicator of cardiac damage from arrhythmia and, perhaps with more in-depth studies, refer to target dosages that could fall within the objective criteria for identifying the anatomical damage of arrhythmia. In our review, the study of heat shock proteins in AF showed unique results in humans and cells but discordant results in animals. The contrast comes from the hypothesis of a protective role of HSP70 in the development of atrial fibrillation. In particular, many studies demonstrate its protective role when intracellular, at high concentrations in patients who do not develop arrhythmia after cardiac injury, such as bypass, and at low concentrations preoperatively in patients who later develop arrhythmia. At the serum level, on the other hand, it becomes an important marker of cellular damage and oxidative stress, also related to increased IL-2 AND IL-4. Could this, like S100β alarmin, also be considered a useful new parameter for stratifying the risk of developing AF? Even more intriguing is the hypothesis of considering AF as an autoimmune disease, if one looks at the results of Korney et al. in which both HSP-70 and antiHSP-70 antibodies were shown to be increased in two cases: (1) in patients with permanent atrial fibrillation; (2) in patients who developed recurrences after catheter ablation. These results can lead us to hypothesize a possible key role not only of inflammation, but also of autoimmunity in the development of AF, triggering the question: what if atrial fibrillation could be an autoimmune disease? Would it open possibilities for hypothetical monoclonal antibody therapies? Finally, if HSP70 is considered as a protective biomarker, novel oral therapies that induce gene expression of the molecule could also be investigated and better explored to protect against the development of arrhythmia, as has already been discussed above in animal models. With these fascinating opportunities, several hypotheses for use could be considered, including inhibition of oxidative stress, evaluation of different mechanisms of action of anticoagulant therapies, genetic studies, and consideration of these alarmins as predictive or prognostic biomarkers of disease onset or severity.

## Figures and Tables

**Figure 1 ijms-23-15946-f001:**
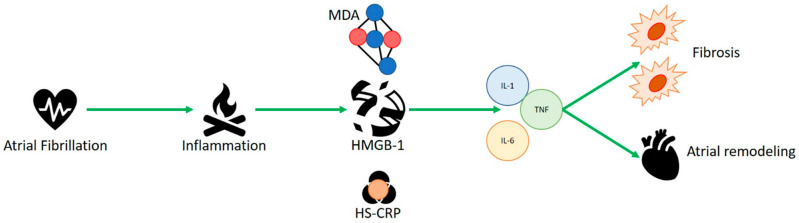
HMGB1 appears to be highly represented in patients with atrial fibrillation, correlating its concentration with malondialdehyde (MDA) and C-reactive protein (HS-CRP), which represents another inflammatory protein. HMBG1 could be amplifier of the inflammatory response with the recall of proinflammatory cytokines such as IL-1, tumor necrosis factor (TNF) and IL-6, which handle atrial remodeling, fibrosis and in the onset of post-surgical AF.

**Figure 2 ijms-23-15946-f002:**
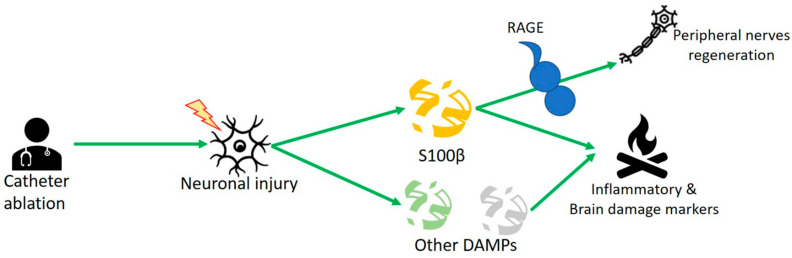
S100β is released because of the neuronal injury of the intrinsic cardiac autonomic nervous system (ICSN), which releases other damage-associated molecular pattern proteins (DAMPs). At the same time, the trophic role of S100β is also shown, which, released from glial cells, through the RAGE receptor would be able to regenerate peripheral cardiac nerves.

**Figure 3 ijms-23-15946-f003:**
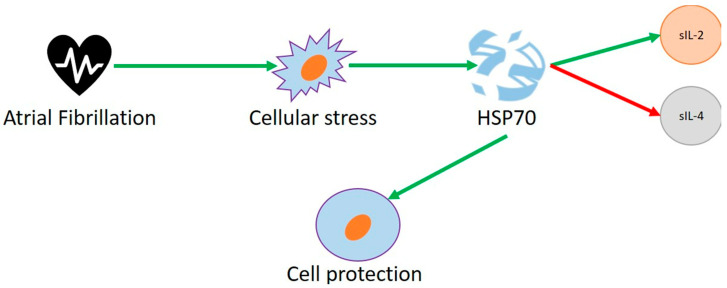
High levels of HSP70 have been shown in patients with atrial fibrillation as a protective antiarrhythmic molecule, correlating with sIL-2 (green arrow) and inversely correlated with sIL-4 concentration (red arrow).

**Table 1 ijms-23-15946-t001:** Studies evaluating S100 proteins.

References	Year	Target
Sramko et al. [25]	2014	Humans
Kato et al. [26]	2016	Humans
Scherschel et al. [27]	2019	Humans
Galenko et al. [28]	2019	Humans
Scherschel et al. [29]	2020	Humans

**Table 2 ijms-23-15946-t002:** Studies evaluating HMGB1 proteins.

References	Year	Target
Yongbo Wu [30]	2013	Humans
Can Qu [31]	2015	Humans
Qiwen Xu et al. [32]	2018	Humans
Kadri Murat Gurses [33]	2018	Humans

**Table 3 ijms-23-15946-t003:** Studies evaluating heat shock proteins.

References	Year	Target
Maurizio Vitadello et al. [34]	2001	Animals
Kyriakos St. Rammos et al. [35]	2002	Humans
Kiriakos Kirmanoglou et al. [36]	2004	Humans
Kaushik Mandal et al. [37]	2005	Humans
Bianca J.J.M. Brundel et al. [38]	2006	Cells
Bianca J.J.M. Brundel et al. [38]	2006	Humans
Masao Sakabe et al. [39]	2007	Animals
Osamu Wakisaka et al. [40]	2007	Cells
Xianqin Zhang et al. [41]	2008	Animals
Mehmet Oc et al. [42]	2008	Humans
Ali R. Afzal et al. [43]	2008	Humans
Jelena Kornej et al. [44]	2013	Humans
Geeta Sapra et al. [45]	2014	Animals
Too Jae Min et al. [46]	2014	Animals
Peili Li et al. [47]	2015	Cells
Can Qu et al. [31]	2015	Humans
Bianca C. Bernardo et al. [48]	2015	Animals
Mikel Allende et al. [49]	2016	Humans
Meng Han et al. [50]	2016	Cells
Marit Wiersma et al. [51]	2017	Cells
Sabine Westphal et al. [52]	2019	Humans
Denise M. S. van Marion et al. [53]	2020	Humans

## Data Availability

Data are taken from previously published articles since this is a review of the current literature to date.

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
