# Peer review of "Alarmins as a Possible Target of Future Therapies for Atrial Fibrillation"

_ijms, 2022, doi:10.3390/ijms232415946_

Round 1

Reviewer 1 Report

The paper by Prof. E. Imbalzano et al. is a nonsystematic review, whose aim was to evaluate the role of specific alarmins in the pathophysiological pattern of atrial fibrillation.

Alarmins are endogenous, constitutively expressed, chemotactic and immune activating proteins/peptides that are released as a result of degranulation, cell injury or death or in response to immune induction, playing important roles as initiators and participants in a diverse range of physiological and pathophysiological processes.

The Authors analyzed in this paper the role of HMGB1, S100 and Heat Shock Protein 70 (HSP70). 

Based on their research, they drew up the following conclusions:

-       HMGB1: this protein seems to be involved in the inflammatory process of AF, being correlated with oxidative stress. At the same time, HMGB1 seems to have a prothrombotic and procoagulant role, potentially involved in atrial thrombosis.

On this basis, the Authors proposed HMGB1 as possible diagnostic and prognostic biomarker of tissue damage and as a potential target for anticoagulant therapy.

-       S-100: The Authors analyzed two members of this large family, S-100b and S-100A4.

o   S-100b seems to have a protective role on the onset of AF and on recurrences of AF after catheter ablation, being higher in patients who experienced fewer recurrences. Therefore, the Authors proposed this protein as possible prognostic marker.

o   S-100A4 seems to be related to the amount of atrial fibrosis, thus being considered as a direct indicator of cardiac damage from arrhythmia. 

-       Heat Shock Protein (HSP): a dual role has been identified for this protein, depending on its location. When intracellular, it seems to have a protective role in the development of AF (being higher in patients who do not develop arrhythmia after cardiac injury, such as bypass, and lower in patients who instead develop AF after cardiac injury). At the serum level, instead, its level is correlated to the oxidative stress and to IL-2 and IL-4 levels, becoming a marker of cellular damage, potentially useful in the risk stratification of AF development.

Although potentially interesting, the results presented in the paper raise several concerns.

1.     Many technical flaws can be identified. The manuscript fell short in providing novel key mechanistic findings and the findings are, in the end, largely correlative and suggestive. 

2.     The whole manuscript needs an English style revision. Many phrases are poorly written and difficult to understand. 

3.     The figures provided, whose role should have been to clarify and summarize what was explained in the text, are actually incapable of fulfilling that function. I would suggest authors revise and improve figures so that the reader can easily follow all the different parts of the study.

4.     There are several critical points in the protein S-100 section. 

a.     At line 436, authors stated “ S100b appears to be increased in circulation in patients with AF after ablation injury or after pulmonary vein isolation surgery. Moreover, its behavior seems to be inversely proportional to recurrence.” This sentence is not in line with two works (39, 41) cited by the authors themselves. In particular, in the paper by Scherschel et al. (41 - DOI: 10.1093/europace/euaa159) even if higher S100B values were associated with freedom from AF after PV-isolation, when patients were stratified in high and low amounts of S100B release, differences in the arrhythmia-free survival (depicted as Kaplan–Meier plot) are only detectable in the CB-PVI group.

b.     In a previous study, Scherschel et al. (39 - DOI: 10.1126/scitranslmed.aav7770) stated that “patients with AF receiving catheter ablation did not show differences in baseline plasma S100b concentration compared with controls without AF”. This is in contrast with the authors’ assumption of a possible protective role of the protein in the onset of AF. 

c.      Sentence in line 184 “Moreover, at six months after surgery, patients had fewer recurrences of AF” is completely decontextualized. It must be reviewed.

d.     Line 197-200 “ A study on 243 patients……no calcium-binding protein B S100b” have no correlation with the rest of the dissertation. It should be better explained or erased.

In the end, the explanation of the role of S100 protein is extremely inaccurate and confusing. It must be thoroughly reviewed. 

Author Response

Dear referee, many thanks for your suggestions, please see attached the response point to point

Round 2:

Manuscript has been largely improved since the first reviewing. I have enclosed few more
comments
and question that may be improved.
Line 51 : what kind of electrical alteration is related to? Please add a reference that explain
electrical
remodelling that induce hypercoagulable state of AF. Are the author talking about fibrosis?
Line 132 133: what means CG, GG, CC please explain.
Line 202 203: meaning of this sentence is unclear. Moreover: fibrosis is a kind of atrial
remodelling.
Does the author talk about the electrical atrial remodeling? (same question than in line 51, please
precise what kind of electrical alteration is it? Effective refractory period modulation? Action
potential duration, automaticity or post depolarization activity?
Line 222: is it the autonomous nerve action potential or cardiomyocyte’s one? Moreover is it a
decreased of action potential duration or amplitude?
Line 224 to 226: meaning is unclear. What type of patients had fewer recurrences of AF. The one
wih
higher S100 or all patients after catheter ablation?
Line 227: cardio-neuro ablation of sympathetic ganglia localized around pulmonary vein are known
to
have benefical effect on arrythmia recurrence after catheter ablation. Cardiac sympathetic
innervation ablation is expected to be associated with AF occurrence as a modulator or the coumel
triangle, by favouring automaticity and modulation of the action potential duration. If S100B is
increased after cardio-neuro ablation it could be a predictive marker of AF non recurrence after
catheter ablation. However it appears only an hypothesis that it could have a protective effect in
AF
recurence. Furthermore, cardio neuroablation is not the first target of AF catheter ablation. Please
add reference that support interest of cardio neuro ablation in AF.
Line 329: ROAF meaning is not precise. Previous AF in this paragraph talk about post operative AF.
It
is recurrence of post operative AF that we are talking about or AF without structural underlying
structural heart disease.
Line 404 -4095 is incorrect and has to be corrected.
Line 48°0: what is precisely the electrical dysfunction and atrial remodeling? It has to be precise.
Line 510: I would be more suggestive. “than HMGB1 can be considered.” We need further studies
to
confirm that. Then I would precise arrhythmia occurrence rather than prognostic factor of
arrhythmia. Or prefer predictive factor rather than prognostic? If prognostic factor refers to the
risk
of thrombo-embolic event, I would suggest to write this at the end of the sentence explaining
relation between HMGB1 and AF.
Line 536: What you mean by anatomical damage of AF: left atrium dilatation, neuronal injury?
Please be precise.

Author Response:
Suggested changed has been incorporated into the manuscript.

Reviewer 2 Report

Brief summary:

Authors review an interesting field of interaction of inflammation and Atrial fibrillation. They particularly described potential role of three different inflammatory proteins HMGB-1, S100 and heat schok proteins. Then they hypothesized different therapeutically strategy that could be tested.

General comments:

I would have like to get more details on how inflammation would induced AF (molecular interaction with trigger of Af and mechanism of maintenance). Strong evidence are published on prothombotic aspect of inflammatory. Less evidence are published on pro-arrhythmic effect of inflammatory. I would have like to get more information on how alarmins interact with triggers of AF and mechanisms of AF maintenance. Hypothesis on AF occurrence and progression of left atrium fibrosis are multiple and chronic inflammatory is only one of them. I would suggest to introduce more genrally the place of inflammation in AF occurrence and maintenance.

Concerning physiopatholgy of AF: approach of the authors sometimes mix underlying AF triggers, mechanism of AF maintenance and thrombogenesis. I would have like to better separate actions of alarmins on different step of AF pathophysiology.

Most of the reference of this rewiew referred to post-operative AF which is a well known entity. Sometimes it is considered as an AF with a provocative factor and is sometimes hypothesized to be related with different underlying mechanism that the most frequent AF without underlying structural heart disease.

I would suggest to separate part 3.1, 3.2, 3.3 in few paragraphs with under title in order to increase readability.

Conclusion is clear but a bit long. I would recommend to add specific conclusion in each paragraph and a more general conclusion.

HMGB-1 and human studies (figure 1)

Brief summary: HMGB-1 is a ubiquitous proteins which modulates cellular stress response and inflammation. Post operative occurrence of AF is associated with an increased expression of G genotype of HMGB-1. Cardiac surgery are known to be associated with pro-inflammatory condition. Thus it is hypothetized by the author that anti-inflamatory treatment may have an antiarrhythmic effect. Detailed interaction with prothrombotic state of AF with HMGB-1 is then described by the author.  Finally, they concluded that HMGB-1 could be an independent predictor of AF.

>>> I would have like to get more explanation on HMGB-1 anti-arrhythmic pathophysiology.

>>> Concerning the pro-thrombotic state: Once again AF could also favoured inflammatory by itself and thus HMGB1 increased could be a consequence of AF increasing burden. Therefore, HMGB1 could be also a marker of AF rather than a predictor. Moreover, accepting their hypothesis we could expect that the thrombotic state of patient would be increased in persistent AF. However, as far as I know, no evidence have been yet published to confirm that. I would have recommend to be modulate the conclusion of this paragraph.

S100 protein and human studies (figure 2)

Brief summary: S100 proteins are important messenger transducers of intracellular C2+ signals. It appears to be related with an antithrombotic state (modulation of the innate immunity), anti-arrhythmic effect, pro-fibrotic action and a biomarker of brain damage. Given this mechanism they listed the multiple possible interaction of S100 and AF.

>> Again, I would have like to get more explanation on S-100 anti-arrhythmic pathophysiology. Ca2+ handling is known to be related with triggers of AF. Does S100 proteins also interact with trigger of AF?

>>> Autonomous nerve system interact with AF. It is not clear here if the authors hypothetized that S100 proteins interact with this autonomous nerve system. Furthermore they don’t precise if it is the vagal or sympathetic cardiac innervation.

Heat Shock Proteins and human studies

Brief summary: HSPs are molecular chaperones that are key for the preservating cellular functions by preventing misfolding and aggregation of polypeptides and facilitating protein folding. Cardioprotective role of HSP70 Is well described. It is then hypothetized and explained how AF could be as an autoimmune disease.

>>> Conclusion on autoimmune aspect of AF in very interesting I would just emphasized that AF mechanism could be multiple and Autoimmune would be one of them.

Specific comments:

Line 18: This causes significant local inflammatory reaction that feeds and sustains the arrhythmia.

>>> how inflammatory could induce AF. Cause or consequence?

Line 41-42: The generation of quick multiple ectopic electrical pulses is able to start and sustain irregular electrical activity of atrial fibrillation.

>>> this physio-pathological mechanism is not well described here and must be referenced.

Line 45: These electrical alterations also help the hypercoagulable state AF-associated.

>>> what electrical alteration are the author talking about? Premature atrial complex? Or fibrosis…

Line 46-47: Failed electrical regularity and excessive ectopic contractility antagonize local atrial hypo contractility, increasing endothelial expression of plasminogen activator inhibitor

>>> meaning? Antagonizes local atrial contractility or explains atrial hypo contractility. It has to be precise.

Line 50 -51: Their common factor is represented by the inflammatory state involved in each of these disorders, which plays a key role in the pathophysiology of AF

>>> inflammatory state is one of the common factors of these disorders, with left atrial enlargement, left atrial fibrosis, adenosinergic system modulation, cardio vascular risk factors, etc….

Line 72-73: Several studies have shown a key role for HMGB1, heat shock proteins and s100 proteins in AF physiopathology.

>>> no reference available here?

Table 1 and 2:No animals or cells target available for those proteins to support the hypothesis?

Line 156-157: Moreover, exactly where the inflammatory mechanism 156 is continuous and constant, as in permanent AF, it seems to occur in higher concentrations 157 than in the paroxysmal AF condition, as an independent predictor of AF.

>>> AF could also favoured inflammatory by itself and thus HMGB1 increased could be a consequence of AF increasing burden. Therefore, HMGB1 could be also a marker of AF rather than a predictor. Moreover, accepting their hypothesis we could expect that the thrombotic state of patient would be increased in persistent AF. However, as far as I know, no evidence have been yet published to confirm that. (Confer genral comments)

Line 180:  What “the murine intracardiac neuros” referred to? What characteristic of the potential action is decreased, Amplitude, duration? Is has to be explaines.

Line 255: it is not clear what “cardioversion” make reference to. I expect that patient who have cardioversion always have recurrence of AF.

Line 434: Therefore, another therapeutic approach could consist precisely in 434 an upcoming HMGB1-blocking anticoagulant therapy.

>>> Does the author recommend the use of HMGB1 as an additive anticoagulation therapy associated with DOA or as a third option of treatment?

Author Response

Dear referee, many thanks for your suiggestions,  please see attached the reponse point to point

Round 2

Reviewer 1 Report

The authors correctly addressed the previous issues and the manuscript has significantly improved